# In Silico Analysis of a GH3 β-Glucosidase from *Microcystis aeruginosa* CACIAM 03

**DOI:** 10.3390/microorganisms11040998

**Published:** 2023-04-11

**Authors:** Gustavo Marques Serra, Andrei Santos Siqueira, Fábio Alberto de Molfetta, Agenor Valadares Santos, Luciana Pereira Xavier

**Affiliations:** 1Laboratório de Biotecnologia de Enzimas e Biotransformações, Instituto de Ciências Biológicas, Universidade Federal do Pará-UFPA, Belém 66075-110, Brazil; 2Laboratório de Tecnologia Biomolecular, Instituto de Ciências Biológicas, Universidade Federal do Pará-UFPA, Belém 66075-110, Brazil; 3Laboratório de Modelagem Molecular, Instituto de Ciências Exatas e Naturais, Universidade Federal do Pará-UFPA, Belém 66075-10, Brazil

**Keywords:** cellulose hydrolysis, β-glucosidase, cyanobacteria, *Microcystis aeruginosa*, comparative modeling, molecular dynamics

## Abstract

Cyanobacteria are rich sources of secondary metabolites and have the potential to be excellent industrial enzyme producers. β-glucosidases are extensively employed in processing biomass degradation as they mediate the most crucial step of bioconversion of cellobiose (CBI), hence controlling the efficiency and global rate of biomass hydrolysis. However, the production and availability of these enzymes derived from cyanobacteria remains limited. In this study, we evaluated the β-glucosidase from *Microcystis aeruginosa* CACIAM 03 (MaBgl3) and its potential for bioconversion of cellulosic biomass by analyzing primary/secondary structures, predicting physicochemical properties, homology modeling, molecular docking, and simulations of molecular dynamics (MD). The results showed that MaBgl3 derives from an N-terminal domain folded as a distorted β-barrel, which contains the conserved His–Asp catalytic dyad often found in glycosylases of the GH3 family. The molecular docking results showed relevant interactions with Asp81, Ala271 and Arg444 residues that contribute to the binding process during MD simulation. Moreover, the MD simulation of the MaBgl3 was stable, shown by analyzing the root mean square deviation (RMSD) values and observing favorable binding free energy in both complexes. In addition, experimental data suggest that MaBgl3 could be a potential enzyme for cellobiose-hydrolyzing degradation.

## 1. Introduction

Cyanobacteria are photosynthetic microorganisms commonly found in aquatic and terrestrial environments. They have different cellular strategies for survival, physiological capacity, and adaptation, which favor their colonization in these habitats [1]. Cyanobacteria are present in several habitats such as fresh water, reservoirs, lakes, and water channels frequently found in Brazil [2]. Eutrophication frequently occurs due to a several factors, including depths, extended retention time, climate changes, and poor sanitation [3]. One of the many freshwater cyanobacteria commonly found in Brazil and responsible for harmful toxic products is *Microcystis aeruginosa*, which is known to be associated with blooms and recognized for producing a wide range of bioactive natural compounds (secondary metabolites) with diverse biological properties, including heptapeptides, that are harmful to human health: microcystins and cyanopeptolins [4,5,6].

Due to the photosynthesis efficiency and the minimal growth requirements that *M. aeruginosa* present, these cyanobacteria are promising enzyme producers. The enzymes reported include lipases, proteases, amylases, laccases, and enzymes involved in carbohydrate accumulation and degradation of lignocellulosic biomass, such as cellulase complexes and hemicellulases [7,8]. β-glucosidase is an interesting enzyme with wide industrial applications from food products to renewable energy production. Furthermore, is has activities in biological systems and against various types of substrates: cellobiose, glycolipids, glucosylceramides, and cyanogenic and flavogenic glycosides have aroused interest in research and use for biotechnology and its processes [9,10].

β-glucosidases are typically found together with endo- and exoglucanase (cellobiohydrolases), where they perform enzymatic hydrolysis of lignocellulosic biomass to produce fermentable sugars [11]. During this process, endoglucanase and exoglucanase activities are inhibited by cellobiose produced as a product of their reactions. The β-glucosidases reduce the inhibition effect by converting the cellobiose to glucose monomers and stimulating the rate of biomass degradation [12]. Thus, this shows the importance of studies of novel β-glucosidases with attractive characteristics for application in industry. β-glucosidases have also been linked to other applications, including in wine production and beverage whitening [13], extraction and biotransformation of active compounds from plants [14,15], food technology in cassava detoxification [16], and other various applications of commercial interest that increase the importance and potential of these enzymes.

According to the work of Henrissat and co-workers [17], β-glucosidases are classified into families of glycosyl hydrolases (GH) in the database of Carbohydrate-Active enZymes (CAZy) using as criteria the coding sequences, similarities between them, and types of folding. In CAZy, β-glucosidases prevail in the GH1 and GH3 families, which are the most commercially relevant and well-studied, but are also present in the GH5, GH9, and GH30. Additionally, we can classify β-glucosidases into three groups according to their substrates and specificity: (I) aryl β-glucosidases; (II) cellobiases, which hydrolyze oligosaccharides only; and (III) wide-range β-glucosidases, which exhibit broad activities on different substrate types [18]. A reliable prediction of the structure, catalysis mechanism, and identification of complex carbohydrates and active site residues is possible using the classification available listed in CAZy.

In this study, a novel β-glucosidase from *Microcystis aeruginosa* CACIAM 03 with high yield was identified and characterized through an in silico approach. Subsequently, the primary/secondary structures were analyzed and their conserved residues were identified. Next, homology modeling revealed the structure as composed of two domains: a C-terminal domain, and an N-terminal domain which contains a conserved catalytic dyad His–Asp residue commonly present in the GH3 family. Moreover, MD simulations and binding free energies provided applicable information regarding orientation and interactions during the binding process, and release products showing that the complex formed is stable throughout the trajectory and is favorably with lower energy in all methods. This study can contribute to the design of cost-effective β-glucosidases for industrial cellulose degradation using the production of those enzymes through the use of cyanobacteria as a biofactory, and contribute to our knowledge.

## 2. Materials and Methods

### 2.1. Organism and Amino Acid Sequence

The *Microcystis aeruginosa* CACIAM 03 strain was isolated from the Tucuruí hydroelectric power station lake (3°49′55″ S and 49°38′50″ W) of the state of Pará (Brazil) and it belongs to the Amazonia Collection of Cyanobacteria and Microalgae (CACIAM*),* with its genome fully sequenced and annotated by the Laboratory of Biomolecular Technology at Universidade Federal do Pará (Belém, Brazil) [19]. The amino acid sequence of its β-glucosidase was retrieved from GenBank under the accession number: OCY13127.1.

### 2.2. In Silico Analysis of Primary and Secondary Structures

Using SMART and CDD methods, the amino acid sequence of β-glucosidase from *M. aeruginosa* CACIAM 03 was analyzed to identify conserved domains, motifs, and signatures [20,21]. For the secondary structures, three distinct techniques and servers were utilized: the GOR IV, SopMA, and CFSSP servers [22,23,24].

### 2.3. Identification of Conserved Motifs

The Proteins Basic Local Alignment Tool (BLASTp) was applied to compare the target protein with other previously sequenced ones from *Microcystis* sp. available at National Center for Biotechnology and Information (NCBI) [25]. The ClustalW tool was used for sequence alignment, and identification was possible using ENDscript [26,27].

### 2.4. ProtParam Analysis

ProtParam tool [28] (SIB Swiss Institute of Bioinformatics, Lausanne, Switzerland) predicted the physicochemical parameters. Parameters including molecular mass (MW), theoretical isoelectric point (pI), extinction coefficient (EC), instability index (II), aliphatic index (AI), grand average of hydropathicity (GRAVY) of the β-glucosidase were provided.

### 2.5. Homology Modeling

In this study, the Protein Data Bank (PDB) [29] database was investigated using BLASTp. The crystal structure of glycoside hydrolase from *Synechococcus* sp. PCC 7002 (PDB ID: 3SQL) was chosen according to alignment score, E-value, and structure resolution. Alignment of the sequences was possible using ClustalW, and MODELLER 10.2 [30] was used to build the structural models. A total of 100 models were generated considering different conformations, ranked by modeler objective function (molpdf) and discrete optimized protein energy (DOPE) score best values.

An automatic refinement loop tool of MODELLER 10.2 was used after model building. The stereochemical quality was evaluated using a Ramachandran plot in the MolProbity server [31], the quality of folding was evaluated using Verify3D [32] and ERRAT [33], and, finally, the root mean square deviation (RMSD) was built between the target and template.

### 2.6. Molecular Docking and Molecular Dynamics Simulations

For the molecular docking and molecular dynamics (MD) simulation, three-dimensional (3D) cellobiose (CBI) and β-D-glucose (BGC) ligands were retrieved from PubChem [34]. Molecular docking was carried out using the DockThor server [35], and for a docking simulation, the size of the grid and discretization was 20 × 20 × 20 and 0.25 Å, respectively, and the grid box was centered in the N-terminal domain binding site.

As the initial conformations for the molecular dynamics (MD) simulations, the docking poses of MaBgl3 with CBI and BGC ligands were used. LigPlot+ [36] was used to analyze the molecular interactions between the docked ligands and the amino acid residues of the enzyme. The protonated states of ionizable residues were determined based on the pKa values predicted using the PDB2PQR server [37] for receptors at pH 4.5. AMBER 18 [38] was used for all MD preparation and production phases, which resulted in the production of 100 ns of MD simulations for both complexes. FB14SB force field [39] was applied to the ligand and protein. To neutralize the charges, Na+ or Cl− counter ions were added. Then, for each system, TIP3P water molecules were placed in a 10 Å cubic box in each direction of the protein [40]. Energy minimization was performed in five steps, four of them using 3000 cycles of steepest descent and 5000 cycles of conjugate gradients for each model; the heavy atoms were restrained by a harmonic potential of 1000 kcal/mol.Å^−2^ [41].

To restrict the heavy atoms, the heating and equilibration procedure was divided into 14 steps. The temperature was gradually increased until 300 K. Langevin dynamics (thermostat) were employed with a collision frequency of 3.0 ps^−1^. A harmonic potential of 25 Kcal/mol.Å^−2^ was employed in the initial steps and turned off during step 13. The heating procedure lasted 650 ps until step 13 and was carried out using an NVT. Subsequently, a 2 ns equilibrium phase was employed in an NPT ensemble. The SHAKE algorithm was employed to restrict the bond vibration of all hydrogen atoms. The particle mesh Ewald method was used to calculate electrostatic interactions, using a cut-off value of 10.0 Å [41]. All graphical molecular representations in this study were generated using the molecular viewer program PyMol [42].

### 2.7. Binding Free Energy Calculation

The binding free energy of the enzyme–ligand complexation (${\Delta G}_{\mathrm{bind}}$) was calculated using the molecular mechanics Poisson–Boltzmann surface area (MM–PBSA) [43], molecular mechanics generalized born surface area (MM–GBSA) [43], and solvated interaction energy (SIE) [44,45] methods. These calculations were based on 5000 snapshots from the last 10 ns of the MD simulation.

MM–PBSA/MM–GBSA methods employ only the unbound and bound states (i.e., endpoints). The binding free energies (${\Delta G}_{\mathrm{bind}}$) can be estimated from the three energies of three reactants according to the following equation:(1)$${\Delta G}_{\mathrm{bind}}={\Delta G}_{\mathrm{complex}}-\left(G_{\mathrm{receptor}}+G_{\mathrm{ligand}}\right)$$
where *G*_complex_ is the free energy for a complex, *G*_receptor_ for a protein and *G*_ligand_ for a ligand. Each free energy is calculated using the following equations:(2)$${\Delta G}_{\mathrm{bind}}=\Delta H-T\Delta S\;\approx \;{\Delta E}_{\mathrm{MM}}+{\Delta G}_{\mathrm{solv}}-T\Delta S$$
(3)$${\Delta E}_{\mathrm{MM}}={\Delta E}_{\mathrm{int}}+{\Delta E}_{\mathrm{ele}}+{\Delta E}_{\mathrm{vdW}}$$
(4)$${\Delta G}_{\mathrm{solv}}={\Delta G}_{\mathrm{PB}/\mathrm{GB}}+{\Delta G}_{\mathrm{non}-\mathrm{polar}}$$

In Equation (2), the term ΔH is the enthalpy, $\Delta G_{\mathrm{solv}}$ is the solvation-free energy, and the TΔS term represents the entropy of the system at temperature T. According to Equation (3), the term $\Delta E_{\mathrm{MM}}$ consists of the energy obtained by molecular mechanics, which contains the sum of multiple terms, such as internal energy ($\Delta E_{\mathrm{int}}$), and electrostatic ($\Delta E_{\mathrm{ele}}$) and van-der-Waals ($\Delta E_{\mathrm{vdW}}$) energies. According to Equation (4), $\Delta G_{\mathrm{solv}}$ is the sum of the electrostatic solvation energy and results from the sum of the polar ($\Delta G_{\mathrm{PB}/\mathrm{GB}}$) and non-polar ($\Delta G_{\mathrm{non}-\mathrm{polar}}$) contributions [43].

The SIE [44,45] incorporates aspects from the linear interaction energy (LIE) and MM–PBSA/GBSA methods. SIE treats the protein–ligand system in atomistic detail, and solvation effects implicitly. The calculation is carried out using following Equation:(5)$${\Delta G}_{\mathrm{bind}}\approx E_{\mathrm{inter}}+{\Delta G}_{\mathrm{desolv}}=\underset{electrostatic}{\underbrace{E_{\mathrm{Inter}}^{\mathrm{Coul}}+{\Delta G}_{\mathrm{desolv}}^{R}}}+\underset{nonpolar}{\underbrace{E_{\mathrm{inter}}^{\mathrm{vdW}}+\Delta G_{\mathrm{desolv}}^{\mathrm{np}}}}$$

In Equation (5), the electrostatic SIE component consists of Coulombic intermolecular interaction energy $(E_{\mathrm{Inter}}^{\mathrm{Coul}}$) and the electrostatic desolvation free energy ($\Delta G_{\mathrm{desolv}}^{R}$) resulting from the change in reaction field energy upon binding. The non-polar SIE component consists of van-der-Waals intermolecular interaction energy ($E_{\mathrm{inter}}^{\mathrm{vdW}}$) and the non-polar desolvation free energy ($\Delta G_{\mathrm{desolv}}^{\mathrm{np}}$) that results from changes in the solute–solvent van-der-Waals interactions and changes required to maintain the solute-size cavity in water. Using a set of 99 protein–ligand complexes, the equation is optimized [44].

### 2.8. Extraction and β-Glucosidase Activity

The *M. aeruginosa* CACIAM 03 strain was cultivated for 45 days in 1000 mL of BG-11 medium and incubated at 25 °C, with 12 h of light (3000 lx intensity) and 12 h in the dark. The crude extract of the cells was centrifuged at 10,000× g for 10 min at 4° C and resuspended with Tris–HCl buffer (100 mM, pH 7.0) containing Roche cOmplete™ protease inhibitor under the same conditions. The pellet was treated with an ultrasonic sonicator with 5 cycles of 10 s at 60% power to lyse the cells completely. Again, centrifugation was performed at 10,000× g for 10 min at 4 °C. Finally, the supernatant was gathered, aliquoted and submitted to the following procedure.

Using *p*-nitrophenyl-β-D-glucopyranoside 20 mM (pNPG) as substrate, the activity of β-glucosidase was measured. The reaction mixture contained 0.125 mL of crude extract, 0.125 mL of pNPG 20 mM, and 0.250 mL of citrate–phosphate buffer (100 mM, pH 5). It was incubated at 35 °C for 30 min. This reaction was stopped by the addition of 0.500 mL of cold 0.2 M Na_2_CO_3_. The activity was detected using the release product: *p*-nitrophenol and measured with a spectrophotometer at an absorbance of 400 nm. One unit of enzyme activity was defined as the amount of enzyme required for the hydrolysis of one μmol of *p*-nitrophenol per minute under assay conditions.

The effect of varying pH ranges on the activity of β-glucosidase was determined using 100 mM of various buffers (citrate buffer pH 3.0–6.0; phosphate buffer pH 6.5–7.5; Tris–HCl buffer pH 8.0–9.0). Following the previous procedure, the reaction, with optimum pH activity, was mixed and incubated at several temperatures (20–50 °C) to determine the optimal temperature. Each test was conducted in triplicate.

## 3. Results

### 3.1. MaBgl3 Amino Acid Sequence

Based on BLASTp results, it was possible to identify additional β-glucosidases from the genus *Microcystis* sp. that shared degrees of identity with the enzyme described in this study. Then, the protein was designated as MaBgl3 (*Microcystis aeruginosa* CACIAM 03 β-glucosidase). The amino acid sequence is annotated in NCBI as OCY13127.1 with a length of 526 amino acids and a molecular mass of little more than 57 kDa, Table 1:

We could identify a presence of the N-terminal domain (Ser7–Ala338, 332 amino acids) and a C-terminal domain (Thr383–Glu513, 131 amino acids) with an E-value of 1.78e^−98^ through the CDD server. The lower E-value, the more significant the corresponding sequence studies. Furthermore, SMART confirmed the protein belongs to the glycoside hydrolase family 3 (GH3).

### 3.2. Identification of Conserved Motifs

The identification of conserved motifs was conducted through BLASTp against PDB since the database has structures characterized and elucidated in terms of their family and catalytic mechanism. Multiple sequence alignment has become critical for identifying conserved regions. The alignment of sequences (Figure 1) was performed with the four best results according to identity, similarity, and resolution parameters of the structures from PDB mentioned from the GH3 family, revealing critical conserved catalytic residues for β-N-acetylglucosaminidase and glycoside hydrolases (including β-glucosidase).

A flexible loop containing an Asp196/His198 dyad was identified by the consensus motif [K–H–F–P–G–H–G–X_4_–**D–**S–**H**] (catalytic dyad highlighted in bold). These residues act as general acid/base, and a second motif identified as IVT**D**A projected to the opposite side of the first motif contains the nucleophilic catalyst Asp270. Therefore, these motifs constitute the enzyme catalytic site of MaBgl3.

### 3.3. Secondary Structure Analysis of MaBgl3

Table 2 displays the secondary structure predicted using GOR IV, SopMA, and CFSSP servers. It exhibited a predominance of α-helix 39.37–42.78%, followed by β-sheet 11.6–14.45%, and loops 39.15–45.82%. The two most common secondary structural elements are the α-helix maintained by hydrogen bonds and the occurrence of structures considered disordered, such as loops, which are segments of a protein that join these secondary elements together.

### 3.4. Prediction of Physicochemical Properties

The physicochemical parameters theoretical isoelectric point (pI), molecular weight (MW), total number of negatively/positively charged residues (R+/R−), extinction coefficient (EC), instability index (II), aliphatic index (AI), and grand average of hydropathicity index (GRAVY) were predicted. The polypeptide chain characterization can provide crucial information regarding the structure and characteristics of this protein. According to a computational study, the MaBgl3 protein has a molecular mass of 57.25 kDa and it is acidic (pI 5.01). The results are closely similar to β-glucosidases of *Microcystis* species, as shown in Table 3.

MaBgl3 might be termed unstable based on this investigation, as its instability index value was 41.45 (>40). Proteins having a value below 40 are classified as stable. The GRAVY is determined as the sum of values all hydrophobic residues divided by the total number of amino acids in the sequence. The GRAVY score for MaBgl3 is −0.004, indicating that aqueous interactions are extremely unlikely.

### 3.5. 3D Model of MaBgl3

Since β-glucosidase from *Microcystis aeruginosa* has no crystallographic structure in PDB, the best template was glycoside hydrolase from *Synechococcus* sp. PCC 7002 (PDB ID: 3SQL) for homology modeling according to BLASTp results, as the E-value = 0, max and total score = 397, and query coverage = 96%. Furthermore, we obtained an identity of 51.57% and a similarity of 73.40%.

The best model, generated from the described parameters, had a molpdf of 2707.7 and a DOPE score of −63,825,601. Additionally, the generated model was evaluated using other servers. The first server was MolProbity which predicts the acceptability of a structure through a Ramachandran plot (Figure 2a). The Ramachandran plot analyzes the quality of the model from the characterization of phi *Φ* and psi *Ψ* angles [46]. The assessment showed that 95.19, 4.81, and 0.0% of the amino acid residues exist in favorable, allowed, and disallowed regions, respectively. Furthermore, Appendix A summarizes the evaluation values.

Another evaluation tool is Verify3D which determines the compatibility between three-dimensional structures and the amino acid sequences [47]. The Verify3D score for the MaBgl3 model is 97.32, which indicates that 97.32% of the amino acid residues in the model had an average 3D–1D score ≥0.2. An ERRAT score greater than 50 is acceptable for high-quality models. The ERRAT score for the MaBgl3 model was 89.08, implying that the backbone and non-bonded conformation were acceptable for this study.

The MaBgl3 structure is composed of two domains. The first one is an N-terminal catalytic domain folded as a distorted TIM barrel (β/α)_8_, similar to the typical folding of β-glucosidase enzymes from GH1 and GH3 families. The domain, which involves the amino acids from Ser7 to Ala338, comprises the conserved loop containing the proposed active residues in Figure 3.

A linker loop involving residues Lys341 to Glu359 is on the opposite side of the enzyme’s active site; it may be associated with the kinetic regulation of substrate access to the catalytic residues and connects the domains. The C-terminal domain comprises amino acids Thr383 to Glu513 consisting of a sandwich of three α-helices, five parallel β-sheets, and three α-helices (Figure 4).

This folding of the C-terminal domain resembles the one found in studies of β-N-acetylglucosaminidases as a 3α/6β/3α-fold type that is reasonably far from the catalytic site of the N-terminal domain [48]. The role of the C-terminal domain is not well established. However, it has been proposed that this domain is acting in the fixation of the substrate, in the correct position, to the active site, in adhesion of carbohydrates from the cell wall of microorganisms [49], and in recognition of substrates, whether this is the case of N-acetylglucosamine (GlcNAg) or specific substrates of β-glucosidases.

### 3.6. Molecular Docking Study

The molecular docking results using the DockThor server are shown in Table 4. The MaBgl3–CBI complex presented the lowest binding free energy value when compared to MaBgl3–BGC. In addition, the intermolecular and electrostatic energies corroborated with the value of the free energies in both systems.

Analysis of Table 5 and Appendix A reveals that MaBgl3–CBI had a higher number of interactions in the active site. The CBI formed hydrogen bond interactions with Trp31, Asp81, and Arg444; in addition, there are hydrophobic interactions with Arg16, Tyr28, Leu53, His198, Ala271, Ile273, and Met274. Both CBI and BGC formed hydrogen bond interactions with Asp81 residues, which can be responsible for the proper interaction of the enzyme with the ligands. The ligands in this study presented important interactions with Asp81, Ala271, and Arg444, those being hydrogen bonds and hydrophobic interactions with residues involved in catalysis and product release.

The interaction of His and Asp residues, called the catalytic dyad, plays an essential role during substrate cleavage in the correct pose of its mechanism in NagZ (β-N-acetylglucosaminidase), such as has been observed in two other studies carried out on members of the GH3 family: NagZ from *Burkholderia cenocepacia* in complex with GlcNAc [50] and NagZ from *Bacillus subtilis* [51]. In contrast to glycosyl hydrolases and β-glucosidase, GH3 NagZ enzymes constitute a substantial departure. These studies demonstrated that the catalytic acid–base is a unique histidine/aspartate dyad that occurs within a flexible loop rather than on separate domains.

### 3.7. MD Simulation Analysis

The MD simulations (100 ns) were carried out to study the stability and conformational behavior of the complex. Then, the root mean square deviation (RMSD) was analyzed; results are shown in Figure 5. The average values for MaBgl3–CBI and MaBgl3–BGC were 2.09 ± 0.34 and 2.82 ± 0.38 Å, respectively. The RMSD of the ligands was 1.58 ± 0.19 and 0.97 ± 0.16 Å for CBI and BGC, respectively. Interestingly, the BGC ligand shifted in the last 20ns, showing some peaks until the 100 ns of simulations.

According to Cheng, RMSD values up to 3.0 Å are acceptable for stable systems during MD simulation [52]. Notably, the MaBgl3–CBI system (Figure 5a) reached equilibrium at 5 ns, demonstrating that the structure and the ligand were stable through the MD simulation. The MaBgl3–BGC system (Figure 5b) reached stability after the first 20 ns. It is worth mentioning that the MaBgl3 structure had a higher value; however, it remained stable throughout the trajectory.

From Figure 6a it is observed in the RMSF analysis that amino acid residues of the MaBgl3 structure behave similarly to both complexes. This behavior is a result of the binding process in the active site; happening in preparation for the transitory phase, where the enzyme increases the mobility of the residues. The average RMSF values for the MaBgl3–CBI and MaBgl3–BGC systems were 1.10 ± 0.60 and 1.08 ± 0.77 Å, respectively. This demonstrates that the values are quite similar.

It can be noted that regions such as Ser18–Leu52 (red), Arg339–Ile360 (green), Ile386–Pro407 (yellow), and Leu505–Val526 (blue) have high levels of fluctuation in both systems, which are loop regions with high flexibility. It is worth noting that conserved residues such as Arg16, His198, Asp270, Ala271, and Arg444 presented high RMSF values, therefore also being located in loops of the active site that are important for proper catalysis and substrate trapping [53].

### 3.8. Binding Free Energy Analysis

The binding free energy calculations were investigated using the last 10 ns of MD simulations for both ligands studied (CBI and BGC). This was necessary to estimate the structural stability, and assess the contributions of individual residues or energy terms through free energy decomposition analysis [54,55,56]. Table 6 displays the energy values of both complexes calculated using the MM–PBSA/MMGB–SA and SIE techniques. Table 6 shows that the formation of complexes was favorable in all three approaches. The MaBgl3–CBI presented a lower energy than the MaBgl3–BGC complex, indicating a higher affinity. Therefore, the three free-energy calculations are consistent with each other, and also corroborate DockThor results.

Analyzing Table 6, the most favorable components are electrostatic energy ($E_{\mathrm{ele}}$) and van-der-Waals interactions ($E_{\mathrm{vdW}}$). The non-polar interactions ($E_{\mathrm{non}-\mathrm{polar}}$) are likewise favorable to binding free energy, but to a lesser extent, while polar solvation energy ($\Delta G_{solv}$) is unfavorable to total free energy.

The energy contributions of each residue for the complexes obtained using the MM–GBSA method are shown in Figure 7. In this investigation, residues contributing to the $\Delta G_{\mathrm{bind}}$ values below −1.0 kcal/mol were considered important to the binding process. The residues that contributed most to the total energy and stability in the MaBgl3–CBI complex were His198, His 186, Arg155, Arg444, and Asp270, which displayed binding free energies of −5.11, −2.99, −1.80, −1.71, and −1.40 kcal/mol, respectively. The per-residue binding free energy of the MaBgl3–BGC complex revealed that the residues Asp270, Arg155, Asp81, His186, Arg444, and Lys185, exhibited binding free energies of −2.90, −1.77, −1.72, −1.55, −1.46, −1.38 kcal/mol, respectively.

In addition, His198 and His186 are the most important residues in the substrate-binding process. In contrast, the most important residue for the MaBgl3–BGC is Asp270, and it is evident that the lack of His198 corroborates the fact that the general acid/base catalyst is far away from the free glucose ring. Thus, residues His198 and Asp270 are highly conserved and perform crucial functions in GH3 family members. Combining MD simulation with a calculation-based method to bind free energies may considerably improve the reliability of both systems [57].

### 3.9. Validation of Enzyme Activity using Lab Analysis

The MaBgl3 enzyme activity (U/mL) was plotted vs. temperature and pH (Figure 8). MaBgl3 exhibited the greatest influence of temperature change on enzyme activity with high activity at 40 °C and with decreasing activity as the temperature increased (Figure 8a). In the pH curve, optimum activity was reported at a pH of 4.5, and it decreased by about 50% as the pH increased (Figure 8b). It is important to note that activity began with greater enzymatic activity at acidic pH levels. The maximum glycolytic activity of 37.19 U/mL could be inferred at pH 4.5 and 40 °C with pNPG.

## 4. Discussion

*Microcystis aeruginosa* is an excellent option for producing cellulolytic systems. The rate-limiting step in the enzymatic hydrolysis of cellulose is mediated by β-glucosidase [12]. Although the activity of β-glucosidase marks the essential point of enzymatic hydrolysis, its concentration in commercially available enzyme cocktails is low and costly, consequently raising the cost of biofuel generation. Finding organisms capable of producing a high concentration of enzymes for effective cellulose hydrolysis has been challenging. Cyanobacteria are regarded as desirable bio-factories for the production of proteins owing to their high photosynthetic efficiency, minimal growth requirements, ease of manipulation, low-cost, and high biomass output. However, nothing about their utilization as a β-glucosidase producer is known.

These findings are the first report to characterize and focus on structural aspects of *Microcystis aeruginosa* β-glucosidase. A single endogenous β-glucosidase from the CAZy family GH3 has been identified. In addition to the knowledge of substrate recognition of β-glucosidase, we suggest a unique binding mechanism for this family of β-glucosidases from the GH families. Previously, Lee and colleagues (2003) [58] revealed a similar binding mechanism for the bifunctional enzymes α-arabinofuronidase and β-xylosidase. *Cellulomonas fimi* was investigated by Mayer (2006) [49], who also suggested that GH3 enzymes have a distinct binding, catalytic process, and studied function of β-glucosidase and β-N-acetylglucosaminidase.

The physicochemical parameters of MaBgl3 were estimated. The isoelectric point (pI) value of 5.01 for MaBgl3 suggests it is an acidic protein. As well as other β-glucosidases from *Microcystis* sp. with pI between 4.96–5.13, these endogenous cyanobacterial enzymes have greater enzymatic activity at acidic pH than others described in the literature. It displayed optimum activity at an acidic pH equivalent to *Aspergillus awamori* (pH 4.5) [59], *Gongronella butleri* (pH 4.5) [60], and a greater tolerance than *Trichoderma atroviridae* TUB F-1505 (pH 6.2) [61] and *Candida peltata* NRRL Y-6888 (pH 5.0) [62]. The experimental investigation verified the presence of a β-glucosidase with an optimum pH of 4.5 at 40 °C, which confirms physicochemical predictions. Bioprocesses using acidic β-glucosidases might limit the development of an unwanted product formed at a higher pH of operation [63].

The ProtParam instability index (II) (Table 3) may be used as a protein primary structure technique for predicting in vivo protein stability. Here, MaBgl3 presented an II of 41.95, which is similar to other *Microcystis* sp. (42.33–43.30). These results show that they are all predicted as unstable, and this prediction indicates that usual purification methods may not be applied, requiring additional attention and specific physicochemical conditions to prevent the loss of enzyme function [28]. The aliphatic index (AI) refers to the volume fraction occupied by a protein’s aliphatic side chains. The AI value affects the potential thermal stability of proteins. MaBgl3 had an AI of 101.73, suggesting that these proteins are thermally stable and have a significant proportion of hydrophobic amino acids.

The analysis of secondary structures (Table 2) reveals the prevalence of protein loops. Typically, they are generally located on the protein’s surface in solvent-exposed areas and often play important roles. They are responsible for protein shape, flexibility, functions, and physicochemical properties. Despite the lack of patterns, loops are not completely random structures. It was observed that those loops not only interconnect secondary structures or domains in MaBgl3, but are also responsible for approximating the catalytic residues and preventing another substrate or ligand from entering the active site [64,65].

As can be observed in the docking results (Table 5), some residues are vital in performing hydrogen and hydrophobic interactions with the residues Asp81, Ala271, Arg444, which are also crucial for the two complexes in the first place. Furthermore, these amino acids are reported and discussed in the MD simulation section, concerning their role in both systems and the free energy calculation.

The analysis of RMSD values (Figure 5) showed that both systems were stable and equilibrated. From Appendix A, the CBI formed hydrogen bonds interacting with Arg16, Arg155, His186, His198, Asp270, Ala271, Phe443, and Arg444 amino acid residues. The catalytic acid/base His198 is localized at the conserved loop (Appendix A), and it is opposite the glycosidic bond with 2.3 Å of distance (Appendix A). The trajectories of MD simulations indicated that these residues ensure the proper location of the substrate inside the enzyme pocket, bringing the total acid/base catalysis closer to substrate cleavage.

For MaBgl3–BGC, the ligand moved away from the binding pocket and relevant catalytic residues mentioned above, as predicted (Appendix A). Moreover, the peaks seen after conclusion of the MaBgl3–BGC trajectory could be interpreted throughout the simulation as a shift and rotation of the BGC. In Appendix A, the catalytic residue His198 was located at a significant distance (13.4 Å) from the BGC. This is reasonable since there is no glycosidic bond to be cleaved. In Appendix A, the BGC ligand was projected out of the protein structure by not forming hydrogen bonds with any residues. At this position, the amino acid residue Arg444 exhibited only hydrophobic interactions.

The binding free energies computed for the MaBgl3–CBI complex (Table 6) were favorable, with values of −25.69, −14.03, and −6.03 kcal/mol for the MM–GBSA, MM–PBSA, and SIE techniques, respectively. Furthermore, the binding free energies for the MaBgl3–BGC complex displayed values of −13.48, −11.71, and −4.42, respectively, from the same three methods. Thus, comparing the binding free energy values in all three methods, shows that the CBI ligand presented the lowest energy values when compared to the BCG. The energy components of the binding free energies are listed in Table 6, and the electrostatic energy ($\Delta E_{\mathrm{ele}}$) showed the most important contributions and its dominant effect in the binding process.

To better understand the individual contribution of each residue for the binding free energy for the lowest $\Delta G_{\mathrm{bind}}$, a residual decomposition analysis of $\Delta G_{\mathrm{bind}}$ was performed using the MM–GBSA approach. Through per-residue decomposition, it became evident that the contribution of basic and acidic residues governs the strength of interaction with the substrate and its reaction product, while non-polar residues appear to contribute less (below −1.0 kcal/mol) to the pocket binding process.

Thus, our results demonstrated that highly conserved residues, such as Trp31, Asp81, His198, Asp270, and Arg444 are positioned in the binding pocket and have lower binding free energy. These data indicate that these residues have a strong affinity to the substrate, which makes the structure more stable during MD simulation. Figure 7 illustrates the distribution of the individual contributions made by each residue throughout the final 10 ns of MD simulations. Five residues produce significant changes in the interaction energy of the MaBgl3–CBI complex, with the Arg155, His186, His198, Asp270, and Arg444 amino acid residues proven to be crucial to ligand binding of the systems.

As we can observe in Appendix A, a slight conformational modification compresses the His198 residue in the conserved loop, which makes a stronger bond and the residue acts as a general base catalyst. In addition, the data indicate that the conserved loop is highly mobile during the glycosyl–enzyme complex trajectory. Therefore, C-terminal domain residues have been revealed to have a low energy value, which is important during the cellobiose substrate binding process to MaBgl3. It is essential to highlight that the catalytic acid/base His198 is far from the ligand, as shown in Appendix A. Thus, this suggests that it activates initially during substrate cleavage and does not participate in product release, as has been documented for other GH3-family enzymes [66,67].

Similar results were previously described in β-N-acetylglucosaminidases; as postulated by Litzinger et al. [68] the GH3 family represents a significant departure from structures of β-glucosidases that have two-domain active sites for enzyme catalysis to occur, with the His–Asp catalytic dyad unique to the flexible loops of enzymes found in this family. Here, we showed that the C-terminal domain also plays a decisive role in the rotation and binding process of the ligand. However, further studies are necessary on other enzymes present in the GH3 family.

In conclusion, the MaBgl3 was predicted as a monomer with good parameters of validation. The structure exhibited a two-domain architecture with a linker loop that connects those domains. MaBgl3 showed the highest structural similarity with 3SQL from Synechococcus sp. Its N-terminal domain has the architecture of an (α/β)8 barrel distorted, similar to other β-glucosidase structures. The catalytic residues at the active site were identified as His198 (general acid/base) and Asp270 (nucleophile). The importance of residues was partially understood through MD simulation and binding free energy calculations, which suggested them as potential contributing factors for the binding process in carbohydrate cleavage. The analysis included in this study could offer new ideas and alternatives for enzyme production and biotechnological application.

## Figures and Tables

**Figure 1 microorganisms-11-00998-f001:**
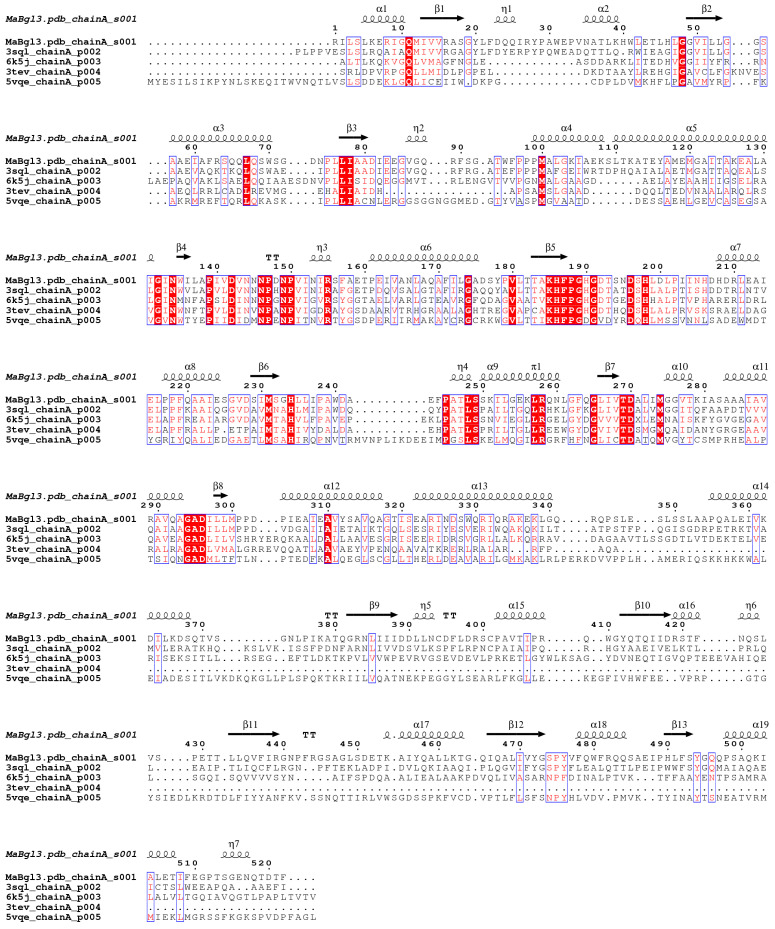
Alignment of the MaBgl3 sequence with selected ones from PDB. (Denoted according to PDB identification and identity percentage PDB ID: 3SQL—glycoside hydrolase from *Synechococcus*, 51.57%; 6K5J—β-N-acetyl-glucosaminidase from *Paenibacilus* sp. str. FPU-7, 39.01%; 3TEV—glycosyl hydrolase from *Deinococcus radiodurans* R1, 38.89%; 5VQE—BglX).

**Figure 2 microorganisms-11-00998-f002:**
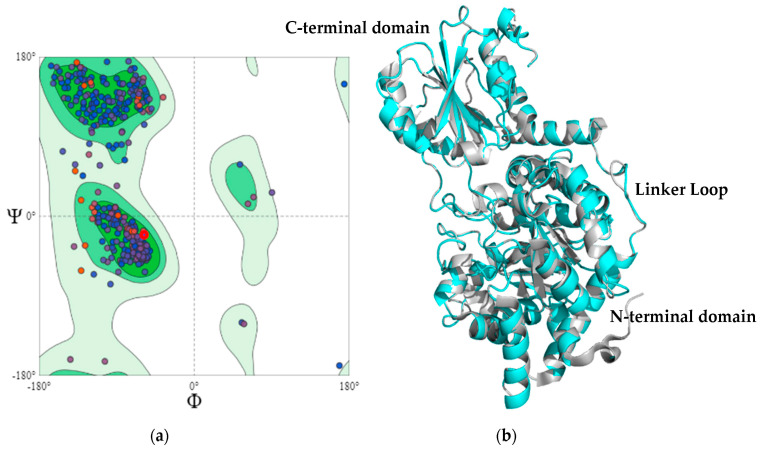
The model validation of MaBgl3 using (**a**) a Ramachandran plot and (**b**) a three-dimensional (3D) structure of MaBgl3 (cyan) in alignment with glycoside hydrolase from *Synechococcus* sp. PCC 7002 (gray).

**Figure 3 microorganisms-11-00998-f003:**
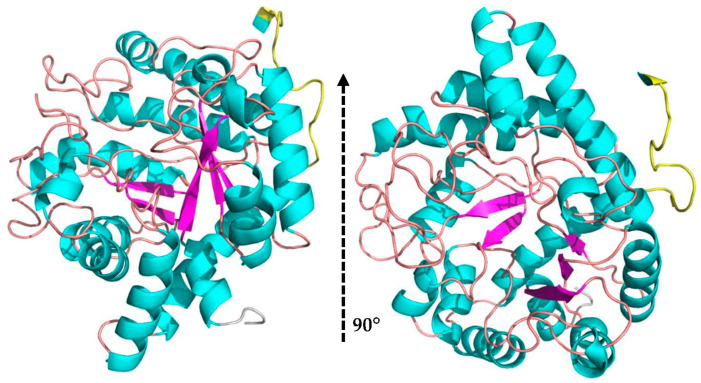
N-terminal domain folded as a distorted TIM barrel in the MaBgl3 model (β/α)_8_ in a side view and upwards view (cartoon structure: α-helices in cyan, β-sheets in magenta, and loops in beige; linker loop in yellow; N-terminal end in grey color).

**Figure 4 microorganisms-11-00998-f004:**
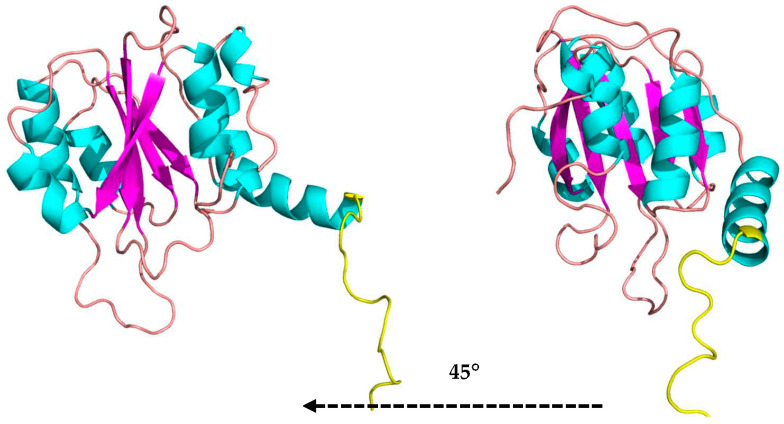
The C-terminal domain folded as a sandwich 3α/6β/3α in two side views (cartoon structure: α-helices in cyan, β-sheets in magenta, and loops in beige) and linker loop in yellow in the MaBgl3 model.

**Figure 5 microorganisms-11-00998-f005:**
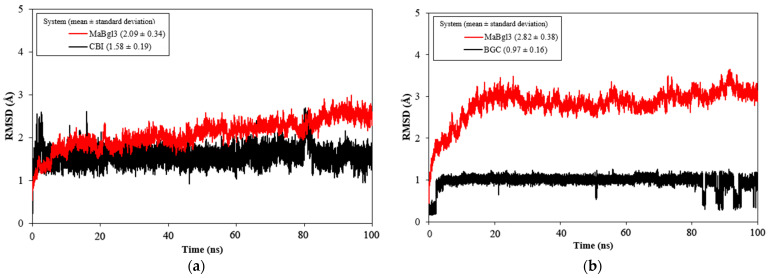
RMSD vs. MD time plot for MaBgl3–CBI (**a**) and MaBgl3–BGC (**b**) systems during 100 ns of MD simulations. The complexes are show in red and the ligands in black. All values are reported in Å.

**Figure 6 microorganisms-11-00998-f006:**
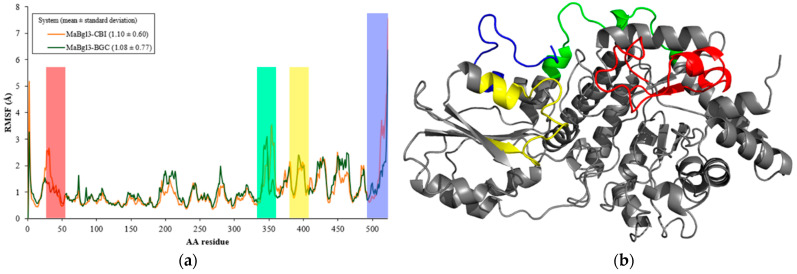
(**a**) RMSF vs. amino acid (AA) residue plot for the main structure of both systems during 100 ns of MD simulations. (**b**) MaBgl3 structure with flexible regions colored and other regions colored in grey.

**Figure 7 microorganisms-11-00998-f007:**
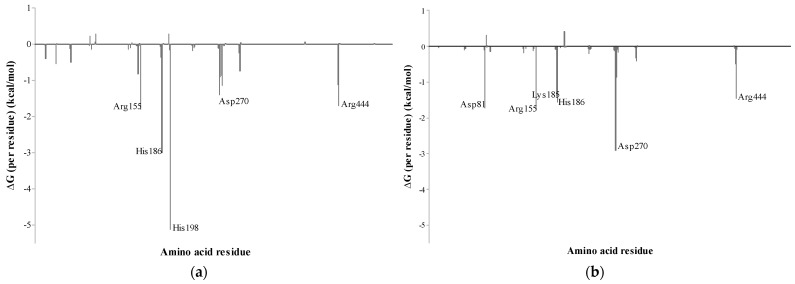
Per-residue binding free energy decomposition of (**a**) MaBgl3–CBI and (**b**) MaBgl3–BGC complexes (in kcal/mol).

**Figure 8 microorganisms-11-00998-f008:**
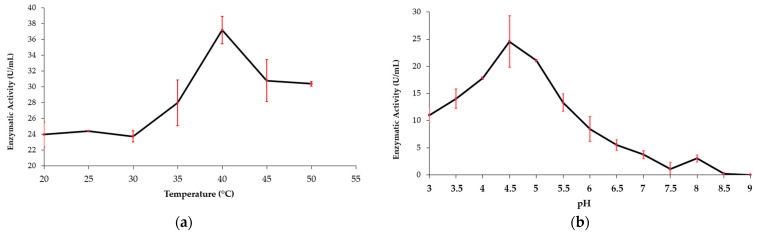
Effects of variation in temperature (**a**) and pH (**b**) on the activity of the MaBgl3 enzyme.

**Table 1 microorganisms-11-00998-t001:** BLASTp result for MaBgl3.

Description	Protein Length (AA)	Molecular Mass (kDa)	Identity (%)	Access Number
β-glycosidase (*Microcystis aeruginosa*) *	526	57.249	100	OCY13127.1
β-glycosidase GH3 (*Microcystis aeruginosa*)	526	57.415	98.10	WP_159250140.1
β-glycosidase (*Microcystis* sp. 0824)	526	57.283	98.10	WP_108937562.1
β-glycosidase (*Microcystis* sp. LEGE 00066)	512	55.728	98.24	WP_051048499.1
β-glycosidase (*Microcystis wesenbergii*)	512	55.750	97.85	WP_199319088.1

* This study; amino acid (AA); kiloDalton (kDa); percentage of identity (%).

**Table 2 microorganisms-11-00998-t002:** Prediction of the secondary structure of MaBgl3 using GOR IV, SopMA, and CFSSP.

	α-Helix		β-Sheet		Loop	
Servers	Residues	(%)	Residues	(%)	Residues	(%)
GOR IV	209	39.4	76	14.5	241	45.8
SopMA	225	42.8	69	13.1	206	39.2
CFSSP	221	42	61	11.6	228	43.3

**Table 3 microorganisms-11-00998-t003:** Physicochemical parameters of β-glucosidases from *Microcystis* sp. using Expasy’s Protparam tool.

Parameters	*M. aeruginosa* CACIAM 03 *	*M. aeruginosa*	*Microcystis* sp. 0824	*Microcystis* sp. LEGE 00066	*M. wesenbergii*
MW (kDa)	57.25	57.42	57.28	55.73	55.75
pI	5.01	5.13	5.13	4.96	5.05
R+	56	55	55	54	53
R-	39	40	40	37	37
EC (M^−1^ cm^−1^)	64.53	71.52	64.40	66,02	65.90
II	41.95	42.56	42.33	42.86	43.30
Stability	Unstable	Unstable	Unstable	Unstable	Unstable
AI	101.73	102.28	102.83	101.46	101.66
GRAVY	−0.004	−0.017	−0.009	−0.004	−0.001

* From this work.

**Table 4 microorganisms-11-00998-t004:** Summary of DockThor results. All results are reported in kcal/mol.

Complex	Binding Free Energy	Total Energy	EvdW ^1^ Energy	Electrostatic Energy	Intermolecular Energy
MaBgl3–CBI	−7.10	24.29	−10.78	−34.85	−45.66
MaBgl3–BGC	−6.18	−3.66	−6.95	−29.88	−36.83

^1^ van-der-Waals energy (EvdW).

**Table 5 microorganisms-11-00998-t005:** Hydrogen and hydrophobic interactions with the amino acid residues of MaBgl3.

Complex	Hydrogen Interactions	Hydrophobic Interaction
MaBgl3–CBI	Trp31, Asp81, Arg444	Arg16, Tyr28, Leu53, His198, Ala271, Ile273, Met274
MaBgl3–BGC	Asp81, Asp270, Phe443	Ile51, Arg155, His198, Ala271, Arg444

**Table 6 microorganisms-11-00998-t006:** Binding free energy and its components are calculated using the MM–GBSA, MM–PBSA, and SIE methods for the MaBgl3–CBI and MaBgl3–BGC complexes. All values are reported in kcal/mol. (Continues).

Components (kcal/mol)	Complexes	
MM–GBSA	MaBgl3–CBI	MaBgl3–BGC
^1^ *E*_vdW_	−23.99 (±3.74)	−10.50 (±2.98)
^2^ *E*_ele_	−80.98 (±9.78)	−64.74 (±6.26)
^3^ *G*_solv_	84.22 (±6.88)	64.80 (±4.20)
^4^ *E*_non-polar_	−4.95 (±0.22)	−3.05 (±0.18)
^5^ Δ*G*_gas_	−104.97 (±8.9)	−75.24 (±5.49)
^6^ Δ*G*_solv_	79.28 (±6.79)	61.76 (±4.16)
^7^ Δ*G*_bind_	−25.69 (±4.69)	−13.48 (±2.89)
MM–PBSA		
^1^ *E*_vdW_	−23.99 (±3.73)	−10.50 (±2.98)
^2^ *E*_ele_	−80.98 (±9.78)	−64.74 (±6.27)
^3^ *G*_solv_	95.58 (±8.96)	66.86 (±4.59)
^4^ *E*_non-polar_	−5.20 (±0.17)	−3.33 (±0.14)
^5^ Δ*G*_gas_	−104.97 (±8.90)	−75.29 (±5.49)
^6^ Δ*G*_solv_	90.38 (±8.92)	63.52 (±4.56)
^7^ Δ*G*_bind_	−14.59 (±5.33)	−11.71 (±3.69)
SIE		
^1^ *E*_vdW_	−24.38 (±3.6)	−10.30 (±3.14)
^8^ *E*_coul_	−35.86 (±4.12)	−28.81 (±2.88)
^9^ Δ*G*^R^	36.35 (±3.13)	28.87 (±2.61)
^10^ Δ*G*_nonpol_	−6.63 (±0.48)	−4.38 (±0.59)
^11^ Δ*G*_pol_	−2.89 (±0.00)	−2.89 (±0.00)
^7^ Δ*G*_bind_	−6.03 (±0.38)	−4.42 (±0.34)

^1^ van-der-Waals energy (*E*_vdW_); ^2^ electostratic energy (*E*_ele_); ^3^ polar contribution (*G*_solv_); ^4^ non-polar contribution (*E*_non-polar_); ^5^ gas-phase free energy (Δ*G*_gas_); ^6^ solvation free energy (Δ*G*_solv_); ^7^ total binding free energy (Δ*G*_bind_); ^8^ intermolecular Coulomb interactions (*E*_coul_); ^9^ reactions energies (Δ*G*^R^); ^10^ non-polar interactions (Δ*G*_nonpol_); ^11^ polar contribution (Δ*G*_pol_).

## Data Availability

Not applicable.

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
