# Peer review of "In Silico Analysis of a GH3 β-Glucosidase from Microcystis aeruginosa CACIAM 03"

_microorganisms, 2023, doi:10.3390/microorganisms11040998_

Round 1

Reviewer 1 Report (New Reviewer)

The manuscript titled "In Silico Analysis of a GH3 beta-glucosidase from Microcystis aeruginosa CACIAM 03" by G. M. Serra et al. is a work on the beta-glucosidase from Microcystis aeruginosa CACIAM 03 to evaluate its potential of cellulosic biomass conversion activity by homology modeling, molecular docking, and molecular dynamics simulations.

I support the paper's publication after they address the following points.

 [1] In the abstract, you mention the importance of the interaction of the three amino acids, Asp81, Ala271, and Arg444; However, there is no direct statement in the text that these three are important. Please add an explanation in the text (discussion section).

  [2] Figure 3 shows the structure of the N-terminal domain of MaBgl3, but it is difficult to grasp its position relative to the overall structure of the MaBgl3 protein. In the figures, please indicate where the N-terminal end and the linker loop to the C-terminal domain are located.

 [3] The values 2.09 +/- 0.34 and 1.58 +/-0.19, shown in line 345, match the value shown in the legend in Figure 5 (a) (MaBgl3 and CBI, respectively). There is a discrepancy in the explanation (line 345); those values are from MaBgl3-CBI and MaBgl3-BCG. Please correct them.

 [4] Please specify the composition of the buffer solution that you are describing as "various buffers (pH 3.0-9.0)" (line 199).

 [5] The temperature and pH dependence of enzyme activity are shown in Figures 8-(a) and 8-(b), respectively; however, it does not match the description in the text (lines 421-423). Please correct one of them. In addition, please indicate the pH condition and temperature conditions for the temperature dependence (Fig. 8-(a)) and the pH dependence (Fig. 8-(b)) of enzyme activity measurements, respectively.

 [6] Please describe the temperature and pH conditions for MaBgl3 enzyme with maximum activity, 37.19 U/ml (line 419) .

 [7] There is no description of the Supplemental videos (Videos S1 and S2). Please add a description of them, the focus site, and the video's time scale.

Author Response

Reviewer 2 Report (New Reviewer)

The manuscript is devoted to computational and experimental studies of the enzyme b-glucosidase, isolated from the cyanobacterium Microcystis aeruginosa CACIAM 03. Overall this is a decent paper. My main objection is that the authors’ claims of the significance of their results are a bit farfetched. As much as I appreciate their methodical work, I highly recommend they tone down the assertive language at the end of the Introduction and Discussion sections. While interesting, their results are still mainly computational models, and as such they at best may potentially contribute to / aid the experimental development of commercially relevant enzymes.

I also have a number of minor questions and remarks:

1. The manuscript is filled with errors and bad formulations with respect to the English language, so I highly recommend it be thoroughly edited by a fluent English speaker.

2. I do not understand the “hence restricting the efficiency and global rate of biomass hydrolysis.” in line 18 in the Abstract. Isn’t the function of the enzyme to facilitate the hydrolysis and not restrict it?

3. The sentence “ Besides, the MD simulations of the MaBgl3-CBI complex was stable by analyzing the root mean square deviation (RMSD) values, and have favorable binding free energy” (lines 26-28) is poorly formulated and needs to be rewritten.

4. The causative connection between the different parts of the sentence in lines 35-37 (the “since” in line 36) eludes me.

5. The lowest energy in line 84 and thereafter is rather “lower” as the authors compare two and not multiple compounds.

6. The “automatic refinement loop by MODELLER 10.2” in line 117 is rather the automatic loop refinement tool of Modeller.

7. I am personally not familiar with the DockThor server, but “ the size of the grid and energy discretization was 20 x 20 x 20 and 0.25 Å” (line 125) sounds wrong, as energy is not measured in distance units.

8. Please, rephrase “Applying the FP14SB force field [38] to the ligand and protein.” in lines 132-133.

9. Please, include the total production simulation time in subsection 2.6.

10. I fail to comprehend the meaning of the sentence “Consequently, forming the catalytic residues of MaBgl3” in lines 236-237.

11. “The results are closely similar to other Microcystis” in line 255 should probably be “The results are closely similar to proteins of other Microcystis species”.

12. How significant are the results for the instability index in lines 260-261, taking into account that both your experimental and MD data show that the enzyme is stable?

13. What are “aqueous interactions” in line 264?

14. I think that the main text of the manuscript would benefit from including a close-up of the catalytic site in Fig. 3.

15. In regard to the sentence “According to Cheng, RMSD values up to 3.0 Å are acceptable for stable systems during MD simulation [53].” (lines 345-346) I would like to note that although RMSD is a normalized quantity, in practice its values still strongly depend on the size of the simulated system. For a 500+ amino acid long protein 3Å RMSD is indeed not quite significant.

16. The BGC RMSD plot in the last 20 ns seems as exploration of two different local minima that correspond to different conformational states of the ligand (lines 351-352).

17. Please, rephrase “According to the RMSF analysis (Fig. 6a) reveals that …” (line 357).

18. I find the interpretation of the RMSF results somewhat problematic. What does “indicate” in line 367 mean? You have a 3D model of the structure, you can check if those residues are indeed buried or solvent-exposed. The lower RMSF regions usually correspond to stable secondary structure elements, such as helices and sheets, while higher RMSF values are characteristic of residues in loops or unstructured/disordered regions.

19. I recommend that “determined” in line 446 be substituted with “predicted” or “estimated”, as this is just an in silico model and not an experimental measurement.

18. The paragraph in lines 515-520 needs rephrasing.

19. The sentence “ Experimentally validated physicochemical investigations revealed relevant and partially proven features” in lines 523-524 is unclear and gives no meaningful information as to what exactly was partially proven or validated.

20. Again, please tone down your claims in lines 84-85 and 526-530.

In summary, the manuscript presents some interesting research. However, in order to merit its publication in the journal Microorganisms, I recommend a thorough revision. 

Author Response

Reviewer 3 Report (New Reviewer)

The manuscript title “In Silico Analysis of a GH3 β-glucosidase from Microcystisae-2 ruginosa CACIAM 03” by Serra et al. described a new beta-glucosidase in light of in silico analysis. The authors described catalytic diad of the enzyme which is a quite rare information for beta-glucosidase. Based on the merit of the manuscript, I think it could be accepted for publication. However, grammatical improvement of the need to be performed. Some issues in the followings may be considered to improve the overall quality of the manuscript before publication.

1.      Lines 46-47, 56-57: need to revise

2.      Lines 61-63, Bgls stands for what? Mention it earlier? Use active form of the sentence

3.      This line “In this study, a novel Microcystis aeruginosa CACIAM 03 with high β-glucosidase 76 yield was identified and characterized through in silico approach.” Is greatly misleading the storey. According to their Methodology [18], the strain was previously published and not reporting in the present study.

4.      Line 244, “Typically, those loops are found on the surface of the protein, responsible for its shape, motion and physicochemical properties [45, 46]” should be transferred to discussion section.

5.      Line 372 “Figure 6. (a) RMSF vs Amino acid (AA) residue plot for main structure of both systems during 100 ns of MD simulations. (b) MaBgl3 structure with flexible regions colored and other regios coloured in grey”. Please edit the typical error regions instead of regios!

6.      How many Beta-glucosidase enzyme are annotated in the Microcystis aeruginosa CACIAM 03 genome? Authors measured Beta-glucosidase enzyme activity of the strain. If multiple enzymes existed in the genome, then how authors could differentiate that which ratios of activity are provided by Bgl3. Sometimes, cellulase also secretes free glucose from oligosaccharide. Therefore, the discussion according to the numbers of cellulase and beta-glucosidase annotated in the Microcystis aeruginosa CACIAM 03 genome, concerning Beta-glucosidase enzyme activity should be incorporated in the manuscript.

Could authors narrate this issue in the discussion section??

7.      Line 496-497, “Thus, it was showed that highly conserved residues, such as Trp31, Asp81, His198, Asp270 and Arg444, that are positioned in the binding pocket and have lower binding free energy”. Further analysis for unveil the catalytic site; authors could perform the virtual comparison of the Bgl3 structure with well-known beta-glucosidase crystal structure from rcs-pdb server and software analysis.

8.      The conclusion section must be improved according to the results and what novelty were observed in the study.

Round 2

Reviewer 2 Report (New Reviewer)

I read the clean version of the revised manuscript, thank you for sending it again! Although I still have some minor objections to the manuscript, I find that it can be published in the present form. I will, however, strongly recommend that you propose to the authors to they take advantage of MDPI's English proof-reading services, as the manuscript is still riddled with minor mistakes and errors

Reviewer 3 Report (New Reviewer)

The authors responded the comments and edited the manuscript. I think it could be accepted for publication. 

This manuscript is a resubmission of an earlier submission. The following is a list of the peer review reports and author responses from that submission.

Round 1

Reviewer 1 Report

The manuscript presents an essentially in silico characterisation of the enzyme β-glucosidase from the cyanobacterium Microcystis aeruginosa CACIAM 03, accompanied by experimentally determined temperature- and pH-activity profiles. The authors use well-established computational methodologies which include homology modelling, docking of a ligand to the enzyme model, simulation of the dynamics of the enzyme-ligand complex and free energy calculations of protein–ligand interactions that should generate meaningful results. However, the language lacks rigor and is often misleading and unintelligible which makes it difficult to fully assess the work done by the authors.

Major issues:

1)  Figure 3 shows a dashed loop. Is this a break in the amino acid sequence? If there is really a break, this may impact the molecular dynamics simulations. Does the homology model contain the full 526 amino acid sequence?

2) The rational for using β-glucoside phosphorylase BglX (PDB ID: 5VQE) as a "reference" protein should be clearly explained (figure 1 and lines 243-247). Moreover, 5VQE contains the ligand 2-deoxy-2-fluoro-alpha-D-glucopyranose (2FGlc), and not beta-D-glucose (BGC), as claimed by the authors (lines 119-120). How does the conformation of this ligand (2FGlc) compares with the pose of BGC in the BglX-BGC complex? An image should shown. Also, the poses resulting from the docking simulations of the complex MaBgl3-BGC should be shown and compared to BglX-BGC. How is the interaction between BCG and the catalytic residues?  What are the atoms in close contact? Is the orientation of the ligand "correct" given what is known about the mechanism? This is important because, at the end of the MD simulations of MaBgl3-BGC, the catalytic dyad (Asp196-His198) was far from the ligand (lines 335-336).

3) Another issue that must be explained is the choice of BGC ligand for docking. Why choose the leaving group? Wouldn't it be more appropriate to study the binding of a substrate, eg cellobiose or the substrate used in section 2.8?

4) In the MD simulations analysis, the authors state that the "MaBgl3 complex proved to be more stable with good correlation with the molecular docking obtained". How can you conclude this? Looking at figure 5, the complex MaBgl3-BGC seems less stable than the complex BglX-BGC. How do you correlate with docking? Or are you referring to the movement of the ligand in the active site?

5) In figure 9, some points lack error bars. Does it mean that there are no replicates for these points? Line 171 states that "All assays were performed in triplicate"

6) What is the meaning of stable/unstable in lines 238-240? How does this compares with the experimental data shown in figure 9.a?

7) The discussion should keep in mind what are predictions and experimental results. The structure of MaBgl3 has not been (experimentally) determined, you have generated a theoretical model. The bifunctional activity (line 397) has not been experimentally tested and is a hypothesis. Finally, i have not found evidence for the last sentence, "we present a new class of β-glucosidase with new properties and mechanisms".

Minor issues:

Line 19: Change "obtained" to "predicted"

Line 20: You should refer the docking calculation, because the MD simulation and free energy calculation that follows are made with a protein-ligand complex

Line 20: "the molecular dynamics" -> "a molecular dynamics simulation"

Line 21: system -> trajectory?

Line 21: binding -> binding to β-D-glucose

Line 24: values of what?

Lines48-51: Please rephrase: the 2 phrases lack a verb.

Line 63: Replace "e" with "and"

Line 66: See comment below for line 90

Lines 72-80: This sentence is huge and hard to follow. Please split it in shorter ones and explain better the reference to the β-N-Acetylglucosaminidases.

Line 82: See comment below for line 90

Line 87: missing "the": annotated by the

Lines 90: "coding sequence" refers to a gene. I believe the analysis is performed on the amino acid sequence.

Line 93: Remove "respectively". The servers are used for the same purpose.

Line 99: Please change this title. You have not performed a physicochemical characterisation. You are making predictions.

Line 102: What is GRAVY? What to do you mean with "recovered"?

Line 104, section 2.5: For the sake of clarity, please refer here what was the template used.

Lines 112-117: Confusing paragraph. Please make it clear what was done with what.

Line 123: Change "The domain N-terminal with" to "The N-terminal domain containing the"

Line 127: "The models generated for target protein and template (5VQE)". Why are you calling 5VQE a template? AFAIK, a template is used homology modelling. And the template used in homology modelling was 3SQL. 

Line 128: You probably mean "to assess" instead of "assessing"

Line 128-129: "The proteins were complexed to a simulated MD of 100 ns" doesn't make sense.

Line 129: "To run these MDs," The PDB2PQR server was not used to run the simulations.

Line 131: Reference [36] should be moved to here.

Line 132: Reference [38] should be moved to here.

Line 132: You should probably remove "respectively", because a single force field was used for both.

Lines 132-133: To neutralize charge, you would need only one of them (Na+ or Cl-). Unless you added both to get a certain concentration.

Line 133: Change "Thus" to "Then"

Line 134: Not clear if you have box of size 10Å or a cubic box extending 10Å from the protein in each direction.

Line 136: Change "to" to "of".

Lines 136-137: Atoms were constrained during energy minimisation? Or was it in the following steps? (check future tense)

Line 138: What do you mean with 14 steps? Temperature jumps? Please clarify.

Line 143: What was the algorithm used to control pressure?

Line 149: born -> Born

Line 150: You probably mean that the last 10 ns of the simulations were analysed with MMPBSA.py script of the AMBER16 software suite.

Lines 155-156: What was "under the same conditions"? I believe the order of events is: discard supernatant, ressuspend pellet, sonicate.

Line 161: What is p-β-NPG?

Line 162: "0.125 mL of 20 mM" of what?

Line 166: What is pNP? What is hydrolized? p-β-NPG or pNP?

Line 170: "maximum temperature"?

Line 174: What do you mean by "consensus sequencing"? In biology, sequencing means to determine the primary structure of something.

Line 180: Table 1. Use decimal point for identity.

Line 182: See comment for line 90.

Line 184: The N-terminal domain, comprising residues 7-338, contains 332 residues, and not 332. The C-terminal domain, comprising residues 383-513, contains 131 residues, and not 127. 

Line 189: What "protein sequence" are you referring to? 

Lines 192-193: See comment for line 174. Figure 1 seems to point to a sequence alignment. Also, what do you mean with "done with four best parameters of identity, similarity and resolution of the structures"?

Line 198: "general" is written twice 

Line 203-204: Please rephrase to make it understandable

Line 205: What "metagenomics library"?

Lines 211-214: What are the %? Identity? Similarity? Why is the % and full name for 5VQE missing? What do you mean with "structure and order"?

Line 221: "they are found" refers to what? alpha-helix and loops? or just loops? Be clear.

Line 224: Reduce the significant digits of the percentages from 4 to 3.

Line 226: see comment for line 99.

Line 235: Table 3: Be consistent with the use (and the meaning) of commas and decimal points and significant digits. Also, what are the parameters R+ and R-? 

Line 256: In figure 2b, the N- and C-terminal domains should be indicated. Also, how does the secondary structure elements in the 3D model compare with what was predicted in section 3.3?

Line 258: "in alignment with Synechococcus sp. PCC 7002". Synechococcus sp. PCC 7002 is a cell. The alignment is with the protein of that cell.

Line 263: I don't see the point of writing about scores after MD here. This should be left to the MD section. The values that are to be commented here are the ones before MD.

Line 267: Rephrase "Presenting a conformation with both domains"

Line 270: Rephrase "lies the full part of the active site Figure 3."

Line 272: Figure 3: Please indicate clearly what is the protein shown. Is it the homology model? Are those two views of the same protein/domain? Also, what are the residues in green and the beige dashed loop in the image on the right? 

Line 274: Where is the loop in the image(s)?

Line 280: Figure 4: see comment for line 272

Line 296: Change "Van" to "van"

Line 300: What is an "active" hydrogen bond?

Lines 304-305: Correct the table caption: the parameters shown do not result from binding between MaBgl3 and BglX (5VQE).

Line 308: "substrate cleavage in the correct pose of its mechanism" is nonsense. Also, what is NagZs?

Line 311: Change simulation to simulations (there are two simulations, right?)

Line 315: Change complex to complexes (there are two complexes, right?)

Line 316: Explain how RMSD is calculated. Main text says "RMSD of the MaBgl3 complex" while the figure legend says "MaBgl3 structure". Is it calculated over protein atoms or over protein+ligand atoms. And over all atoms, or a subset of atoms such as heavy atoms or main chain? Also, how is the ligand RMSD calculated?

Line 329: What is the "C1 carboxyl"? From table 6, it seems that should be the Cdelta...

Line 332: "The active site" of MaBgl3?

Line 333: What is the Nag3 motif?

Line 337: Table 6. Clarify what Table 6 is showing: distances between protein and ligand; in "Atoms", the 1st atom is from the protein, the 2nd atom from the ligand. Then, what "average" distances are those? Aren't they taken from the last MD snapshot? Also, for the sake of clarity, for 5VQE, indicate (eg in parenthesis) what the corresponding residue in MaBgl3.

Line 339: the β-glycosidase is from Microcystis aeruginosa CACIAM 03, not from MaBgl3. 

Line 342: "the protein 5VQE complex showed to be close to the analyzed ligand" is nonsense.

Line 343: Asp288 corresponds to Asp270 in MaBgl3? The same for His206.

Line 362: Use decimal points.

Line 366: Incomplete sentence: each residue of?

Line 367: Are you referring to the complex MaBgl3-BGC?

Line 370: What is loop-13? It includes Phe443, but not Arg444?

Line 383: What do you mean by "higher tolerance"?

Line 391: Not clear to what "These findings" refers too.

Line 404-408: This paragraph needs to be rewritten to be meaningful.

Line 418: MaBgl3 complex -> MaBgl3-BGC complex

Line 420: Change "pNPG" to "p-β-NPG"

Line 458: Reference is incomplete.

Author Response

We appreciate the reviewer for taking time to carefully review the manuscript and give detailed and constructive comments, which has greatly helped to improve this paper. Below is our point-by-point response to each comment. Please see the attachment.

Reviewer 2 Report

In their manuscript 'In Silico Analysis of a GH3 beta-glucosidase from
Microcystis aeruginosa CACIAM 03' the authors aim at the computer-assisted characterization of the respective enzyme from cyanobacteria, especially to "elucidate its structure and mechanism of catalysis" (cited from Abstract text). They apply a number of standard tools for 2D analysis, build a homology model, perform two short molecular dynamics (MD) simulations, and experimentally measure the pH dependency of the enzyme. Despite this seemingly complete characterization, the manuscript is not suitable for publication in Microorganisms in its present form.

There are two major issues with the work:
1, The aims stated in the Abstract are not reached: Neither the authors
provide the reader with a validated structural model to be of further use, nor
do they elucidate the mechanism of catalysis on the atomic level.
2, The length of the single MD simulations is too short to guarantee convergence; additional MD simulations would be needed to confirm the findings from the simulations and minimize the probability of simulation artifacts.

Thus, the authors should address these two issues prior to a resubmission:
- At least one additional MD simulation for each system has to be performed and analysed to confirm the findings. In order to link simulations and experimental findings, a second set of simulations at elevated temperature (e.g. 40 °C) would further help characterizing the system (should still be stable).
- The aim of elucidation the enzymes mechanism or the mode of action should be removed from the abstract text, and experimental and theoretical methods and findings joined in a better way.
- An electronic supplement is needed for this work with at least the structural models of the simulated systems, e.g. the last snapshot from the MD as pdb-file (including hydrogens and ligand).

However, the manuscript has further minor issues, that should also be
addressed in a revised version (not-complete list):
- BGC ligand: Why was the *2D* ligand needed, what used for?
- Add force field and its reference for ligand description (Glycam?).
- Please comment on the large harmonic potential of 1000 kcal/(mol*A^2) to
constrain the atoms.
- Please add reference for MM/GBSA method.
- Table 1: please correct typo "glicosidase" -> "glycosidase"
- The authors should explain, why they used different structures from the PDB as template for homology modelling and as MD reference system (3SQL vs 5VQE)!
- Figure 2b: The authors may want to add descriptions for parts of the
protein, which are also used in the manuscript text (e.g. domains, active site
etc.)
- Figure 3+4: Please add note that a single structure is shown in two
orientations, and provide a description of the differernt structural
representations (what is shown in sticks/ribbon, in yellow/blue ...) and
provide the sequence numbers of the domains shown...
- Residues in the binding pocket could be added to section 3.6.
- Energy values (kcal/mol) in Tables must not be given with 3-4 figures after
the decimal *dot*, since this suggests an accuracy of the methods, which is
not reached.
- For which atoms was the RMSD value computed for? All atoms, heavy atoms only?
- Table 6 provides "average distance [...] at the end of the simulation": this
is a contradiction. However, the key interaction pairs should be analyzed over the course of the simulation (Figures -> Supplement) and mean as well as standard deviation should be given in the Table.
- The authors perform a binding analysis via MM/GBSA; the denote the energies as "free energies" (Delta_G), although the do not include any entropic contribution into the final energy. -> rename the energies, include entropies, or comment on the omission of the entropy contribution and its impliction on Free Energies.
- Figure 6+7: Must be improved! No black background, representations that are better understandable, all shown residues with names.
- Figure 8 shows the decomposition of the binding energies. Maybe the authors also want to perform some distance analysis over the trajectories for those interactions to see, whether all of those are stable interactions or
fluctuations exist.
- The effect of pH dependency should be elucidated more clearly by inclusion of atomistic interaction schemes, where the influence of (de)protonation can be seen. Maybe a chemical reaction scheme with the molecular mode of action depicted would be helpful. Otherwise, the pH dependency appears as completely unlinked result in the list of properties with molecular detail.
- The authors may want to rename the MD reference system in the manuscript text and figures/tables: The pure PDB code is not correct, since the system had beed added hydrogens, water molecules, counter ions etc.
- Video files (-> supplement) showing the flexibility of the systems would
also help the reader to comprehend the system.

Author Response

(The authors gave the same response as above.)

Reviewer 3 Report

The manuscript submitted by Gustavo Marques Serra et al was reviewed. A molecular dynamics study of the complex between beta-D-glucose and a Beta-glucosidase from GH3 family in cyanobacteria was carried out. Furthermore, the PH-dependent feature of enzyme activity was experimentally measured.

There are several concerns to be answered before this manuscript is accepted.

1. My biggest concern is how effective a docking study using beta-D-glucose and beta-glucosidase is to reveal enzymic functions. Docking study and MD simulation are effective to reveal molecular interaction between ligand and macromolecules. Focus on molecular recognition before the catalytic reaction: hydrolysis, docking, and consecutive MD simulations must be conducted at least using a repeating unit of cellulose (dimer). Or, am I missing something?

2. Is there any relation between the sections of enzymic activity and in-silico simulation? If not, perhaps the experimental section could be omitted for readers to easily follow the story of the manuscript.

3. Evaluation of delta G values from two methods is missing. The evaluated delta G values by the methods were not close at all. How can we interpret the binding free energy? Some experimentally measured affinity values, e.g., pKi or IC50 are necessary to understand the simulation results, otherwise, it is hard to evaluate.

Minor points:

There are some typos and grammatical mistakes found in the manuscript: for example, in the abstract: “The MaBgl3 model revealed been a monomeric protein with…”. Thoroughly proofreading of the manuscript would be recommended.

Author Response

(The authors gave the same response as above.)

Reviewer 4 Report

The research paper of Serra and co-authors is intended for the special issue of Microorganisms ‘Advances in Microcystis aeruginosa’. Authors performed the comprehensive bioinformatic analysis and molecular modeling of new ferment β-glucosidase, obtained from the cyanobacterium M. aeruginosa

Authors used molecular docking and molecular dynamics simulations to demonstrate the structure of an active site.

Additionally, they extracted ferment from the M. aeruginosa and measured its β-glucosidase activity. This part is really shortened, and the results obtained are not compared with those obtained in bioinformatical studies. Discussion is mainly repeating of the results. Better it would be if authors would speak more about evolutionary differences of different enzimes. 

The English language is OK. The figures can be improved (see below). After fixing all issues, this paper would represent a nice contribution to the special issue.

General Issues:

Figures 3 and 4 – it is unclear what is the difference between two views of each Ð¡- and N-terminal domains. Better draw the arrow and indicate the rotation degree (if there was a rotation). Also, in Fig 4 images are not aligned to each other.

In Figure 8 the subscriptions are too small.  Especially, the free binding energy values.

Minor issue:

Line 156 – which protease inhibitor was used? It should be specified with the name of manufacturer.

Author Response

We appreciate the reviewer for taking time to carefully review the manuscript and give detailed and constructive comments, which has greatly helped to improve this paper. Below is our point-by-point response to each comment. Please see the attachment

Round 2

Reviewer 3 Report

I appreciate the authors' effort to correct the manuscript according to my comments and other reviewers' comments. The docking study with cellobiose has now been included, a more reasonable assumption for the simulation study of the glucosidase enzyme. However, in my opinion, one primary concern remains even in the current form of the manuscript.

In the response letter from the authors, they merely explained the difference between the delta G values by PBSA and GBSA from a methodological point of view. My concern is the validity (reliability) of the calculated delta G values. If the calculated delta G values were not trustworthy, the entire discussion based on the energy values becomes nothing but speculation. Thus, I recommended the authors look for measured equilibrium constant values (e.g. pKi) for this target. Furthermore, the revised delta-G value for MaBgl3-CBI using GBSA is -25 kcal/mol, which is exceptionally low. I have never seen such a strong molecular interaction. How would the authors rationalize the values and discussion? These two points regarding delta-G should be clarified before this manuscript is accepted.